# The Role of Pedagogical Agents in Personalised Adaptive Learning: A Review

**Ufuoma Chima Apoki** [1,*], **Aqeel M. Ali Hussein** [2], **Humam K. Majeed Al-Chalabi** [2], **Costin Badica** [2] **and Mihai L. Mocanu** [2]

1   Faculty of Computer Science, Alexandru Ioan Cuza University, 700506 Iasi, Romania
2   Faculty of Automatics, Computer Science and Electronics, University of Craiova, 200440 Craiova, Romania; aqeel.hussein.it@gmail.com (A.M.A.H.); hemoomajeed@gmail.com (H.K.M.A.-C.); costin.badica@edu.ucv.ro (C.B.); mihai.mocanu@edu.ucv.ro (M.L.M.)
*   Correspondence: ufuomaapoki@gmail.com; Tel.: +40-726-989-772

**Abstract:** Personalised adaptive learning is becoming increasingly popular as a method of providing each student on an online platform with learning experiences that are tailored to their own requirements and peculiarities. This enables learners to go along many learning routes with the shared objective of information and skill development. In such systems, adaptivity and intelligence play distinct roles, with adaptivity being a more data-driven decision-making approach and intelligence being the emulation of human traits in a learning setting. Pedagogical agents, as defined in the field of artificial intelligence, are virtual characters with anthropomorphic features that are introduced for educational reasons. Because e-learning is a continuously growing area, the responsibilities of pedagogical agents change based on the goals that have been established for them. This article provides a systematic evaluation of pedagogical agents' research and empirical data in e-learning from 2015 to 2022. Their responsibilities will be examined specifically in terms of flexibility and variety, realistic simulation, and their influence on learning: performance improvement, improved motivation, and engagement. The article finishes with a discussion and recommendations on pedagogical agents' future directions in this ever-changing world of individualised adaptive e-learning.

**Keywords:** pedagogical agents; personalised adaptive learning; adaptivity; intelligence; e-learning; systematic literature review

## 1. Introduction

Personalised adaptive learning (PAL) is a new pedagogical approach that extends beyond standard e-learning and classroom-based learning and is facilitated by smart learning technologies. The paradigm encompasses elements such as e-learning, m-learning, u-learning (learning anywhere and at any time), and s-learning (smart learning) [1]. This tackles the difficulties of dealing with a dynamic and digital learning environment where learning can be both asynchronous and synchronous with learners from various backgrounds and with varying needs. Based on their definitions and purposes, the two pillars of PAL, personalised learning and adaptive learning, have different beginnings. Personalised learning pedagogy entails a learner-centred teaching approach that is optimised to fit the needs of all learners with the purpose of encouraging individual growth [2,3]. Adaptive learning, on the other hand, incorporates learning scenarios that include technology that monitors students' progress and dynamically adjusts instruction or content to improve educational experience, performance, and engagement [4,5].

PAL, at its heart, integrates the aims of adaptive and personalised learning by taking into account the following elements: individual differences, individual performance, and adaptive adjustment [1]. Personalised learning may occur both within and outside of an online learning environment, whereas adaptive learning necessitates the usage of technology in online learning environments. The growth of big data technology may also

be related to the development of PAL: the generation and storage of data in more detailed ways and at faster paces.

Brusilovsky and Peylo [6] proposed a class of systems that they termed "Adaptive and Intelligent Web-based Systems". These systems integrate adaptivity and intelligence to give learners personalised experiences that take into account their requirements, performances, and peculiarities. These concepts are commonly used interchangeably in personalised adaptive learning systems (PALS); nevertheless, adaptivity in a system does not always imply intelligence, and vice versa. Adaptivity, which has its roots in adaptive hypermedia systems, delivers diverse learning experiences and reactions based on a learner's choices and requirements [6,7]. Intelligence, which has its roots in Intelligent Tutoring Systems, often gives the same level of help and tutoring to all learners [6,7]. In practice, there is no apparent separation between these technologies, and successful e-learning applications combine both intelligence and adaptability in responding to student peculiarities and demands.

One major problem in e-learning settings is the high student-to-teacher ratio, which vastly outnumbers that of classroom-based learning environments. The use of pedagogical agents derived from the field of artificial intelligence in e-learning is an attempt to account for this. Software agents, which include properties such as longevity, semi-autonomy, proactivity, and adaptability, have been assigned to fulfil the roles of mentors, tutors, peers, etc. in learning environments as early as the 1970s [8,9].

This paper summarises the findings of a systematic literature review on the functions of pedagogical agents in customised adaptive learning systems in this setting. Based on Kitchenham and Charters [10], the review method evaluates research in papers published between 2015 and 2022 with the following objectives:

**RO1:** Define the responsibilities of pedagogical agents in personalised adaptive learning systems.

**RO2:** Investigate the impact of incorporating pedagogical agents into e-learning environments.

The rest of this study is organised as follows: Section 2 presents some related reviews of pedagogical agents in e-learning, Section 3 explains the methodology for this systematic review, Section 4 examines the main theories and claims of the roles and effects of pedagogical agents in PAL, Section 5 presents the results and a discussion of the roles and effects of pedagogical agents and our recommendations, and Section 6 concludes the review.

## 2. Related Reviews on Pedagogical Agents in Personalised Adaptive Learning

The use of software agents in e-learning settings has long been an intriguing study issue for a variety of reasons, and multiple studies have explored it from various viewpoints. One apparent reason is the capacity to manage and automate work for a large number of users, as acquired in an e-learning situation, duties that would be hard to do by humans. Agents may also imitate intelligence and, as a result, can play various roles in various learning contexts. A lecturer in Mahmood and Ferneley [11]'s study wrote:

> "If the agents can do half of my routine tasks, that would be nice... as an academic, I can spend that valuable spare time for more productive activities. I don't mind if those routine tasks are handled by agents."

Mahmood and Ferneley [11]'s study includes data gathered over a 12-month period through participant observation and group discussion. The study's goal was to provide an empirically based paradigm for developing and evaluating animated agents in e-learning settings. They specifically investigated the roles that animated agents may effectively adopt, as well as whether social and cultural aspects impact the interaction between learners and animated agents. The research also looked at when, how, and for whom the agent roles were established.

Veletsianos and Russell [9]'s study on pedagogical agents examined assertions about agents' educational functions in publications released between 2005 and 2011. These assertions include adaptability and variety, the capacity to provide realistic simulation,

the ability to accommodate learners' socio-cultural demands, and the ability to encourage engagement, motivation, and accountability. The promises also include improved learning experiences and performance. Their findings show that the actual data supporting such statements is frequently varied and, in some cases, conflicting. Long-term studies of the outcomes of educational agent intervention and deployment in naturalistic scenarios and open-ended environments were recommended in the study.

Kim [12]'s research looked at how educational agents may serve as interface companions, socio-cognitive aids, interaction partners, and social models. The author also investigated the development of pedagogical agents in terms of the instructional role and personal characteristics. Agent competency, interaction type, emotion, gender, ethnicity, multiplicity, and feedback were mentioned as components to incorporate in effective agent design.

Martha and Santoso [13]'s research looked at publications spanning 2007 to 2017 that focused on how the design of educational agents affects learning settings. The form of communication, such as text or voice, and the look of the agent are among the design considerations taken into account (2D, 3D, or human). Facial expressions, gender, ethnicity, and emotion are also influences in look. These outward appearances and functions, such as guide, teacher, mentor, motivator, and so on, are then compared to dependent variables like learning outcomes, learning behaviour, and agent value. Based on their findings, they believe that mixed design variables have the potential to dramatically increase learning performance and learner behaviour.

A more recent study by Papoutsi and Rangoussi [14], which looked at papers from 2009 to 2019, focused on pedagogical agents in e-learning. First, they examined the level of interest and context trends in the use of software agents throughout the time period. They also looked at the possibility of interaction between teaching agents and students. Another topic investigated was the pedagogical agent modality: animated/non-animated, speech, text, picture, and so on, as well as the learning results produced by including pedagogical agents in learning situations.

This research is unique in this setting since it focuses on the functions of agents and the expected results when they are included into PALS. Personalised adaptive learning refers to systems that are not totally ITS or AHS but give some type of customisation and adaptivity in real-time based on the learner's choices and knowledge. Furthermore, the pedagogical agents under consideration for this review are those used in e-learning situations.

## 3. Research Methodology

We used the systematic review procedure to collect relevant information from reliable publications for this research. We followed the instructions suggested by [10], which include the following steps for planning and carrying out the research. The stages are depicted at a high level in Figure 1.

### 3.1. Planning the Review

The first part entailed organising the review, which included establishing the objectives, as stated in the Introduction. To summarise, the following are the goals of this study:

1. To define the roles pedagogical agents play in PAL systems.
2. To investigate the projected outcomes of including pedagogical agents in PAL environments.

Specifying the RQs

With the objectives in mind, the following research questions were developed to enable us to obtain and analyse data from primary studies.

**RQ1:** What adaptive and intelligent roles do pedagogical agents play in personalised adaptive learning systems?

This study examines the many roles that pedagogical agents play in giving advice, mentoring, or support during the learning process to address this research question. We'll be looking at the roles from two perspectives: adaptive roles and intelligent roles. Adaptivity-

focused roles are more data-driven and involve direct guidance/navigation, adaptive display, and information filtering. Tutoring, collaborative learning, and intelligent monitoring are examples of intelligent roles that replicate either a mentor or a peer.

**RQ2:** What are the outcomes of incorporating pedagogical agents in learning environments?

To answer this research question, we examine the impacts that researchers want to achieve by including agents into learning settings. We'll explore three outcomes. The first will be the traditional enhanced performance of each individual learner. Another will be the ability to complete assigned academic tasks. The final will be better engagement, motivation, and increased learner accountability.

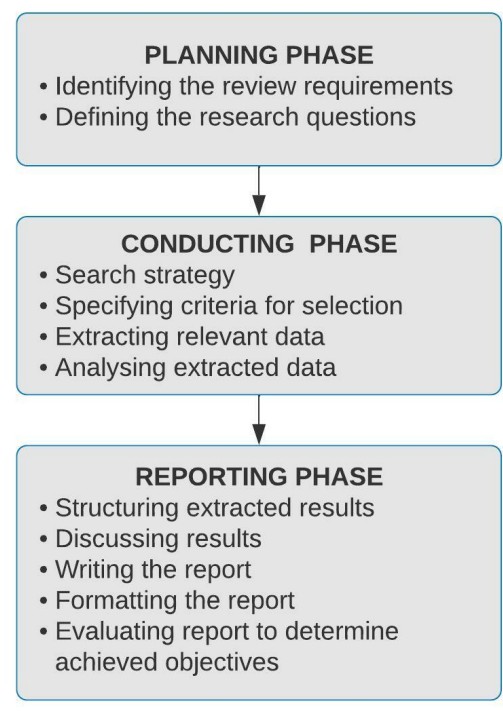

**Figure 1.** The SLR process.

*3.2. Conducting the Review*

This stage of the review process entailed defining the search strategy as well as the criteria for choosing the main studies that were eventually included in this evaluation. Our extraction and analysis procedures are also detailed briefly.

3.2.1. Search Strategy

Following the aims of this study and developing the RQs, the search method for this review was developed. The search structure was developed and the search was carried out in order to narrow down the amount of articles to those that would be relevant to our purpose. The keyword set was derived from comparable studies that were utilised to conduct reviews that met the aims of this study. Synonyms for the set of keywords were manually discovered in order to broaden the search results. For the year-span chosen for the review, the keywords were rotated with different combinations.

The first set of keywords includes "software agents", "pedagogical agents", and "artificial intelligence", which were combined with another set of keywords including "personalised adaptive learning", "adaptive hypermedia", "intelligent tutoring", "e-learning", "online learning", "virtual learning", and "adaptivity".

The following online databases were searched: Google Scholar, Science Direct, IEEE Xplore, Research Gate, SpringerLink, and ACM Portal. To keep up with new research, publications were limited to the years 2015 to 2022. The list of articles was organised

and ordered using Google Spreadsheet, and duplicates were removed automatically and confirmed manually.

### 3.2.2. Criteria for Study Selection

A number of articles were obtained using the list of keywords given in the search strategy. The following criteria were used to decide which studies were included and which were omitted from the review.

Criteria for inclusion:

- Articles published between 2015 and 2022
- Articles written in English
- Articles that performed some form of real-time personalisation during learning according to the learner's preferences
- Articles that incorporated pedagogical agents in the personalisation of learning
- Articles that appeared in conference proceedings and scholarly journals

Criteria for exclusion:

- Books
- PowerPoint presentations or publications that just include abstracts
- Articles with inaccessible texts
- Articles in which the agent roles are not clearly established
- Articles that lack real-time personalisation

## 4. Agent Theories in Personalised Adaptive Learning

Since its inception into the field of artificial intelligence, software agents have been described in many ways. Some definitions that distinguish them from simple computer programs are as follows:

*"an entity that functions continuously and autonomously in an environment in which other processes take place and other agents exist."* [15]

*"an encapsulated computer system that is situated in some environment and that is capable of flexible, autonomous action in that environment in order to meet its design objectives."* [16]

*"a system situated within and a part of an environment that senses that environment and acts on it, in pursuit of its own agenda and so as to effect what it senses in the future."* [17]

The following are some essential characteristics of software agents:

- Autonomy: the ability to exert some control over their behaviour and internal condition without the need for direct intervention.
- Reactivity: the capacity to detect changes in their surroundings and respond appropriately.
- Pro-activity: the ability to not just be reactive but also to engage in goal-directed behaviour
- Continuity: the capacity to run constantly or just when necessary,
- Social capacity: the ability to engage with other agents (through agent-communication language in a multi-agent system architecture) and humans (through natural language).

While the qualities listed above are not exhaustive, Franklin and Graesser [17] emphasise that a software agent should meet the first four requirements.

Pedagogical agents are software agents embedded in learning contexts to accomplish a variety of educational aims. The origins of pedagogical agents may be traced back to 1970s research on Intelligent Tutoring Systems (ITS). For example, they typically serve as teachers or motivators and can communicate with students by gestures, natural language, or facial expressions. Because they may provide cognitive assistance to the student [18] as well as social enrichment to the learning experience [19], pedagogical agents are widely implemented in online learning environments. For example, by addressing questions,

agents may give human-like support and lessen student fear and frustration by seeming welcome and kind.

Numerous assertions have been made in several research works and study reports about the functions pedagogical agents can play in learning situations and the results of their inclusion. We'll go through the ideas that surround these roles and consequences in a quick overview.

Pedagogical agents have been widely reported to be flexible and versatile. Through flexibility, support, and scaffolded guidance, research on pedagogical agents argue that they can enhance learning, provide knowledge, and support both cognitive processing and metacognitive skills. Furthermore, according to experts, pedagogical agents have the ability to monitor and adjust to students preferences in learning and educational background in order to personalise and adapt instruction. Agents can give intelligent scaffolding to learners by providing suitable challenges or information by employing adaptive systems that are configured to respond to users intelligently. Agents, in essence, observe the behaviour of learners to determine when they might want assistance and then deliver help and guidance promptly.

Another point is that pedagogical agents produce realistic simulations by mimicking human behaviour. Agents may, for example, demonstrate procedural tasks, employ gestures and gaze as instructional tactics, play out think-aloud scenarios to imitate reasoning and metacognition, and model acceptable social behaviour to demonstrate how people behave. Agents serve as actors, models, simulators, and manipulatives in digital learning environments. Furthermore, researchers anticipate that pedagogical robots might enhance the credibility of simulations by including a virtual body and engaging with learners in a realistic manner.

Researchers have also stated that by offering possibilities for social contact, agents can meet a wide range of learners' socio-cultural demands in virtual settings. Agents, for example, can function as peer learners and collaborate with people in collaborative tasks if they have the necessary abilities and domain knowledge. Virtual agents, as activity partners, may reduce learner anxiety and increase student empathy by giving peer support, acting as role models, and allowing students to watch mistakes made by the agent throughout the learning process.

Pedagogical agents are used in e-learning settings to fill functions that would typically be filled by humans in classroom-based learning. These responsibilities might include functioning as a peer, tutor, or mentor. To go deeper into these responsibilities, we'll look at adaptivity and intelligence in PALS. Adaptivity in a system refers to a system's capacity to change its behaviour in response to learner demands and other factors [7]. Intelligence, on the other hand, refers to approaches derived from Artificial Intelligence that are used to assist learners in PALS [6]. Figure 2 depicts modern technologies that provide adaptivity and intelligence in PALS.

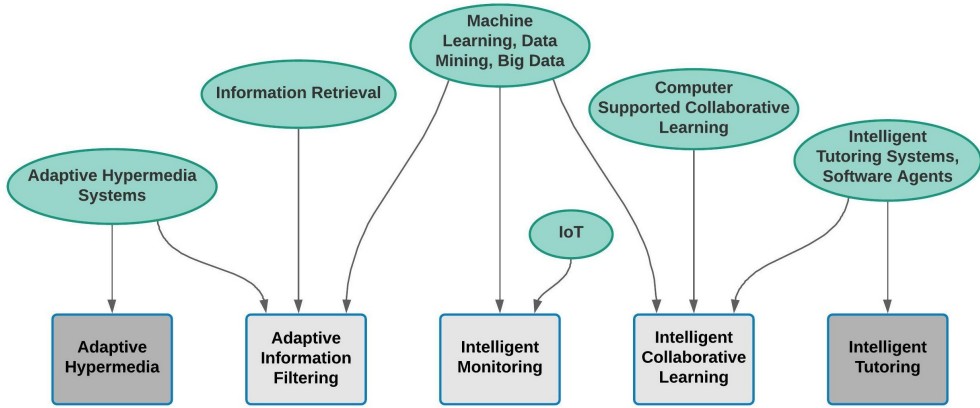

**Figure 2.** Modern technologies of adaptivity and intelligence in PALS.

Table 1 summarises intelligent and adaptive technologies incorporated in PALS [6].

**Table 1.** Adaptivity and Intelligence in Personalised Adaptive Learning Systems.

| | Technologies | Description |
|---|---|---|
| Adaptivity | Presentation | The adaptive presentation technology, which is derived from the field of adaptive hypermedia, delivers a non-static website that dynamically adapts to fit the learner model's objectives, knowledge, and other preferences [20]. |
| | Navigation | Adaptive navigation is a technique that is also associated with adaptive hypermedia. Adaptive navigation provides assistance in an educational hyperspace by modifying the look of visible hyperlinks. The primary aim is to provide the ideal path across the learning space based on the learner model's needs, preferences, and goals [20]. |
| | Information Filtering | Information filtering arose from the field of information retrieval, and it assists a user in finding relevant information in a large body of information. In scenarios such as web searches, results are generated by sorting and filtering information based on the user's choices. Adaptive information filtering can be either content-based or collaborative. |
| Intelligence | Monitoring | The lack of feedback from learners makes it difficult for remote teachers to customise their instructions to the learners' requirements in e-learning. Using artificial intelligence approaches, intelligent monitoring technologies assist the remote teacher in keeping track of the learner's reactions. Intelligent monitoring, which primarily employs data mining and machine learning, attempts to give teacher support in e-learning environments [6]. |
| | Collaborative Learning | Before the internet, collaborative learning technology was formed by combining Computer-Supported Collaborative Learning (CSCL) with Intelligent Tutoring Systems [6,20]. The goal of artificial intelligence approaches is to improve learning experiences through collaborative strategies such as group creation, peer aid, collaboration support, and virtual students. |
| | Tutoring | Intelligent tutoring, which was originally used in ITS, tries to assist the student in the learning process by utilising artificial intelligence. Curriculum sequencing, intelligent solution analysis, and problem-solving assistance are all examples of help [6]. |

## 5. Discussions

This report describes the conclusions based on the examination of the published study. Non-statistical approaches were employed to analyse and interpret the findings. Based on the preceding search method, 32 papers were chosen from the results based on the inclusion criteria. However, only 25 articles were approved for review. Table 2 summarises the final list of primary studies utilised in this review, including the year of publication, design type, and location of publishing. According to the Table, 56% of the studies had designs that included some type of implementation, including testing with learners. The remainder were conceptual ideas with execution plans for the future. Figure 3 depicts the distribution of our studies according to the publishers. As the picture shows, the bulk of the research were published by Elsevier and Springer. MDPI, IEEE, Inderscience, and ACM include other notable publishers.

Table 3 summarises the roles and expected outcomes of the incorporating pedagogical agents in the studies examined. The findings are subsequently presented according to the research questions.

Addressing RQ1: What adaptive and intelligent roles do pedagogical agents play in personalised adaptive learning systems?

**Table 2.** List of Primary Studies of Pedagogical Agents in PALS.

| Year | Ref. | Design | Publication |
|------|------|--------|-------------|
| 2015 | [21] | Empirical | Knowledge-Based Systems |
| | [22] | Conceptual | International Journal of Knowledge and Learning |
| | [23] | Empirical | International Journal of Adaptive and Innovative Systems |
| | [24] | Empirical | Journal of Computer Assisted Learning |
| | [25] | Empirical | Computers in Human Behavior |
| 2017 | [26] | Conceptual | International Journal of Computer Applications |
| | [27] | Empirical | Intelligent Automation & Soft Computing |
| | [28] | Conceptual | Computational Science and Its Applications—ICCSA 2017 |
| 2018 | [29] | Empirical | Interservice/Industry Training, Simulation, and Education Conference |
| | [30] | Empirical | International Journal of STEM Education |
| | [31] | Conceptual | International Conference on Computational Science and Its Applications |
| | [32] | Conceptual | International Journal of Smart Education and Urban Society (IJSEUS) |
| 2019 | [33] | Conceptual | Computers in Human Behavior |
| | [34] | Conceptual | XVIII International Conference on Data Science and Intelligent Analysis of Information |
| | [35] | Conceptual | Heliyon |
| | [36] | Empirical | 2019 CHI Conference on Human Factors in Computing Systems |
| 2020 | [37] | Empirical | International Journal of Emerging Technology in Learning |
| | [38] | Conceptual | International Journal of Electrical and Computer Engineering |
| 2021 | [39] | Empirical | Computers |
| | [40] | Empirical | International Conference on Enterprise Information Systems |
| | [41] | Empirical | Australasian Journal of Educational Technology |
| | [42] | Conceptual | International Conference on Computational Science and Its Applications |
| | [43] | Conceptual | Proceedings - Frontiers in Education Conference, FIE |
| | [44] | Empirical | Asia-Pacific Web (APWeb) and Web-Age Information Management (WAIM) Joint International Conference on Web and Big Data |
| 2022 | [45] | Empirical | Computers and Education: Artificial Intelligence |

**Table 3.** Agents Roles and Outcomes in Surveyed Studies *.

| Ref. | Agent Roles | | | | | | Outcomes | | |
|------|-------------|---|---|---|---|---|----------|---|---|
| | Adaptive Roles | | | Intelligent Roles | | | | | |
| | **P** | **N** | **IF** | **M** | **CL** | **T** | **P** | **TC** | **E/M/R** |
| [21] | ✗ | ✗ | ✗ | ✗ | ✗ | ✓ | ✓ | ✗ | ✗ |
| [22] | ✗ | ✓ | ✗ | ✗ | ✗ | ✓ | ✓ | ✗ | ✓ |
| [23] | ✗ | ✗ | ✗ | ✗ | ✗ | ✓ | ✓ | ✓ | ✗ |
| [24] | ✗ | ✓ | ✗ | ✗ | ✗ | ✓ | ✓ | ✓ | ✗ |
| [25] | ✗ | ✓ | ✗ | ✓ | ✓ | ✓ | ✓ | ✓ | ✓ |
| [26] | ✗ | ✗ | ✗ | ✓ | ✓ | ✓ | ✓ | ✓ | ✓ |
| [27] | ✗ | ✗ | ✗ | ✗ | ✗ | ✓ | ✓ | ✗ | ✗ |
| [28] | ✗ | ✗ | ✗ | ✗ | ✗ | ✓ | ✓ | ✗ | ✗ |
| [29] | ✗ | ✗ | ✗ | ✗ | ✓ | ✓ | ✗ | ✓ | ✓ |
| [30] | ✗ | ✗ | ✗ | ✗ | ✓ | ✓ | ✗ | ✓ | ✓ |
| [31] | ✗ | ✗ | ✗ | ✗ | ✗ | ✓ | ✓ | ✗ | ✗ |
| [32] | ✗ | ✗ | ✗ | ✗ | ✗ | ✓ | ✓ | ✗ | ✗ |
| [33] | ✓ | ✗ | ✓ | ✓ | ✗ | ✓ | ✓ | ✗ | ✓ |
| [34] | ✗ | ✗ | ✗ | ✗ | ✗ | ✓ | ✓ | ✓ | ✗ |
| [35] | ✗ | ✗ | ✗ | ✗ | ✗ | ✓ | ✓ | ✗ | ✗ |
| [36] | ✗ | ✗ | ✗ | ✗ | ✓ | ✗ | ✗ | ✓ | ✓ |
| [37] | ✗ | ✓ | ✗ | ✓ | ✗ | ✓ | ✓ | ✗ | ✗ |
| [38] | ✗ | ✓ | ✗ | ✓ | ✗ | ✓ | ✓ | ✗ | ✗ |
| [39] | ✗ | ✓ | ✗ | ✗ | ✗ | ✓ | ✓ | ✗ | ✗ |
| [40] | ✓ | ✗ | ✗ | ✗ | ✓ | ✗ | ✗ | ✗ | ✓ |
| [41] | ✗ | ✗ | ✓ | ✗ | ✓ | ✓ | ✓ | ✓ | ✗ |
| [42] | ✗ | ✗ | ✗ | ✗ | ✗ | ✓ | ✓ | ✗ | ✓ |
| [43] | ✗ | ✗ | ✗ | ✗ | ✗ | ✓ | ✓ | ✓ | ✗ |
| [44] | ✗ | ✗ | ✗ | ✗ | ✗ | ✓ | ✓ | ✗ | ✗ |
| [45] | ✗ | ✓ | ✗ | ✗ | ✗ | ✓ | ✗ | ✓ | ✗ |

* Adaptive Roles (P: Presentation, N: Navigation, IF: Information Filtering); Intelligent Roles (M: Monitoring; CL: Collaborative Learning, T: Tutoring); Outcomes (P: Performance, TC: Task Completion, E/M/R: Engagement/Motivation/Responsibility). A ✓ represents a feature that was represented in a study while an ✗ represents a feature that was not represented.

### 5.1. Adaptive Roles

As can be seen from Table 3, the adaptive roles of educational agents were not broadly explored as were the intelligent roles in the studies evaluated. Only 2 studies used pedagogical agents to dynamically modify the learning environment's presentation. Agents performing some type of adaptive navigation were included in 7 of the studies. We were only able to determine 2 studies that included pedagogical agents to filter information from a wide pool of knowledge for learners as related to adaptive information filtering. Figure 4 represents the distribution of adaptive roles in terms of the number of publications per year. In the following subsections, we will examine each of the roles using examples from the primary studies that were examined.

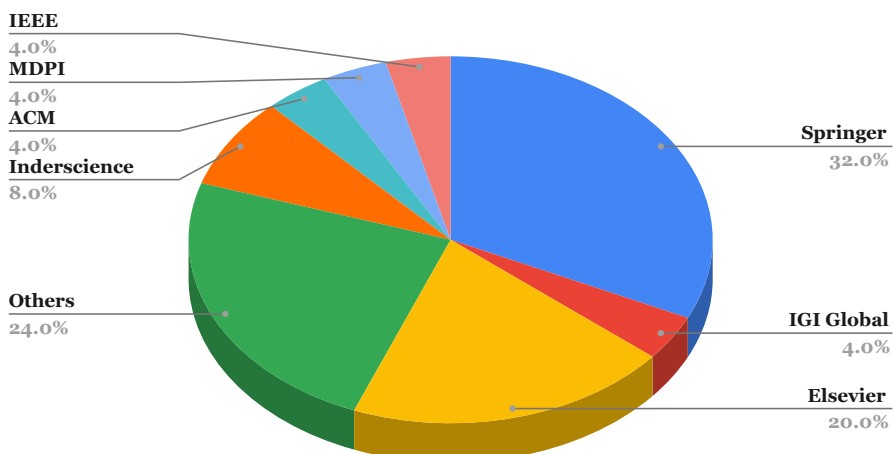

**Figure 3.** Overview of article publishers.

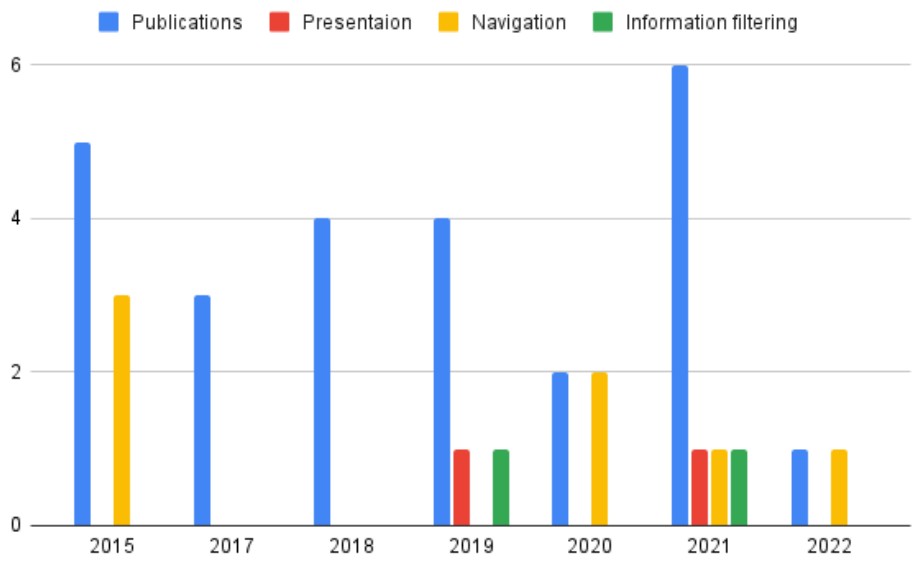

**Figure 4.** Distribution of adaptive roles across the examined period of time.

5.1.1. Adaptive Presentation

Adaptive presentation in an educational hypermedia space includes adaptive multimedia presentation, adaptive text presentation, and adaptation of modality [46]. Inserting or removing fragments, modifying fragments, stretching text, and sorting or dimming fragments are some instances of adaptive text presentation. Hammami et al. [33] propose a MAS architecture for deaf students in their study, with numerous agents doing diverse tasks. A deaf student agent, in particular, is charged with dynamically updating the interface with information communicated from agents in the underlying MAS architecture. The other article in this category used an interface to encourage participation based on the learner's profile and decrease dropouts [40].

5.1.2. Adaptive Navigation

In terms of adaptivity, navigation in an educational hyperspace entails changing the appearance of connections using methods such as direct guidance ("next"/"previous"), adaptive sequencing, link concealment and removal, and adaptive link annotation [47]. The provision of direct guidance is the equivalence of classic ITS curriculum sequencing. The studies in which agents played some sort of navigation function focused on providing

learners with direct guidance [22,24,25,37–39,45]. The design by Apoki [39], on the other hand, also includes link removal.

### 5.1.3. Adaptive Information Filtering

Adaptive information filtering is represented by technologies such as content-based filtering and collaborative filtering [6]. While the goal of content-based filtering is to match the proper information to a student based on his or her preferences and requirements, the goal of collaborative filtering is to connect learners who are interested in the same sort of content. The first instance of a pedagogical agent engaging in adaptive information filtering is the study by Hammami et al. [33], in which the agents in the MAS architecture fulfil the job of obtaining appropriate content for deaf pupils. Their next project will be to leverage YouTube as a resource base for implementation. The second example of adaptive information filtering involves a conversational bot that recommends video fragments to learners depending on their answers to questions in a Massive Open Online Course (MOOC) [41]. These video clips are intended to help students better comprehend the idea associated with the question on which they did not get right. Video clips are also retrieved from YouTube.

### 5.2. Intelligent Roles

The pedagogical responsibilities of agents representing intelligence for the investigated studies are illustrated in Figure 5. Though intelligent pedagogical agent roles were more prevalent than adaptive roles, the majority of studies (92%) had pedagogical agents play roles to complement some type of tutoring activities. Intelligent collaborative learning was integrated by pedagogical agents in 28% of the studies. Intelligent monitoring was also underrepresented, with just 20% of pedagogical agents monitoring and intervening during the learning process. Subsequent subsections will detail each role with some examples from the reviewed studies.

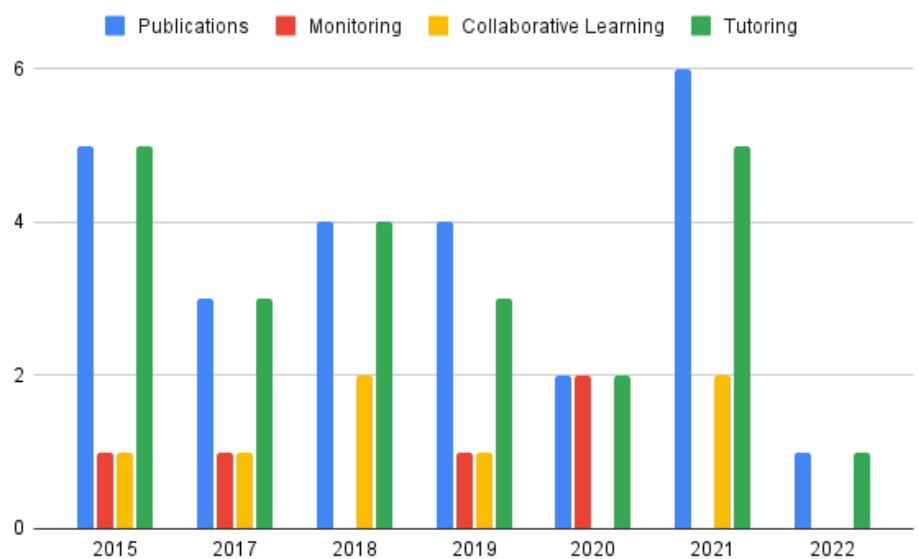

**Figure 5.** Distribution of intelligent roles across the examined period of time.

### 5.2.1. Intelligent Tutoring

Because software agents and ITS are both based on artificial intelligence, there were more pedagogical responsibilities in the intelligent tutoring category [6,7]. Curriculum sequencing, problem-solving support, and intelligent solution analysis are examples of classical intelligent tutoring technology [6].

Curriculum sequencing, which was the function played by the majority of the agents in the studies examined, included activities that offer learners with the most appropriate planned sequence of subjects to learn and tasks to complete (such as examples, questions, and exercises). Multiple agents serve many responsibilities in systems that use agents for curriculum sequencing, such as gathering information to construct or update the learner model and sending such information to other agents whose duty it is to organise content given to the learner.

When a learner is in the process of solving an problem or carrying out a task, pedagogical agents that provide problem-solving support create an interactive help environment. These can take the form of delivering hints or feedback [23–25,30] during the task or suggesting the next steps in solving the problem.

Intelligent analysis of exercise solutions provided by pedagogical agents goes beyond simply presenting bad or right responses. They can analyse the solution and point out any missing or erroneous information. A good example is given in [41] where a conversational agent analyses the responses that learners in a MOOC are supposed to offer and proposes video segments that cover certain ideas that are not correctly addressed.

### 5.2.2. Intelligent Monitoring

Several studies indicated that their designs facilitated monitoring during the review process. However, we distinguish between intelligent class monitoring and pedagogical agents tracking learning activities and updating learner models (as obtained in tutoring activities for the purpose of curriculum sequencing). According to Brusilovsky [20], intelligent class monitoring entails looking for mismatches in a group of learners. Such anomalies are detected as a group of learners who require more assistance/attention than others, and intervention is necessary and supplied in real-time. Mismatches can be represented by dissatisfaction, delayed or quick advancement, and so on. The PAOLE (Pedagogical Agent for Online Learning Environments) agent [26], which senses dissatisfaction and gives some type of motivation to reduce dropouts in MOOCs, is one example in the review.

### 5.2.3. Intelligent Collaborative Learning

Adaptive group creation and peer assistance, adaptive collaboration support, and virtual students are examples of pedagogical agent roles supporting collaborative learning. Several adaptive learning platforms are attempting to add virtual (and visual) students who can interact with learners via voice [30] and text [29,36] as agent design improves. The model by Duffy and Azevedo [25] features four pedagogical agents: *Gavin*, a guide who provides learners with system knowledge and assists them in exploring the learning environment, *Pam*, a planner, is responsible for assisting learners in creating objectives; *Mary*, a monitor, is responsible for ensuring learners' knowledge throughout learning sessions; and *Sam*, a strategist, is responsible for promoting the application of learning methodologies.

### 5.3. Expected Outcomes

Addressing RQ2: What are the outcomes of incorporating pedagogical agents in learning environments?

Figure 6 depicts the distribution of projected outcomes of integrating pedagogical agents in relation to the number of publications in each year. Improved performance was by far the most represented in the research for the expected outcomes by integrating pedagogical agents (80%). 44% of the outcomes were to get learners to complete some sort of academic task with the agents providing scaffolding and hints. 36% of projected outcomes involved some sort of enhanced student engagement, motivation to learn and reduce dropout, and an increased sense of responsibility while learning.

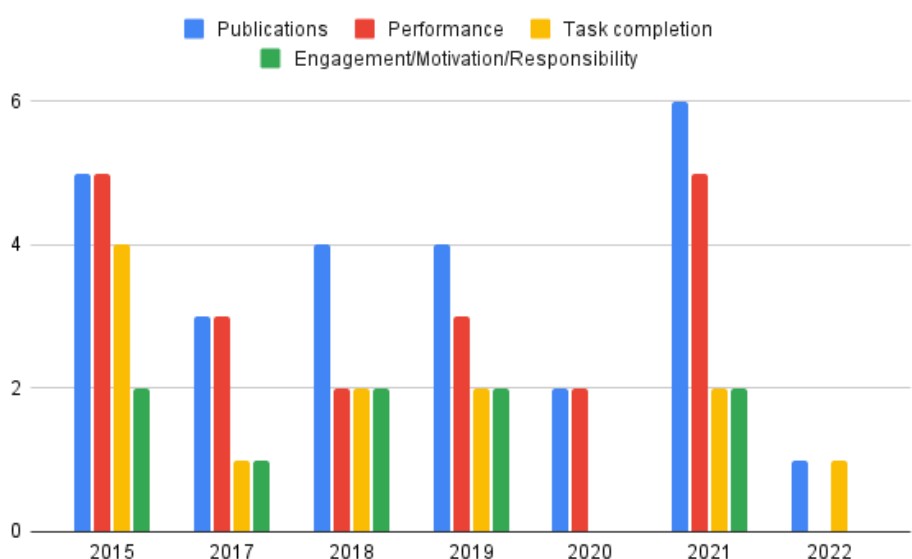

**Figure 6.** Distribution of expected outcomes across the examined period of time.

In the next subsections, we'll look at each of the outcomes: improved performance, task completion, and increased engagement, motivation, and responsibility, using examples from the studies we reviewed.

5.3.1. Improved Performance

One of the recurring arguments in Veletsianos and Russell [9]'s review is that pedagogical agents boost learner efficiency with respect to classical knowledge dissemination. The majority of activities involving enhanced performance are associated with tutoring-based agent roles accomplished via improving understanding, memory, recall, problem-solving, self-efficacy, and knowledge transfer.

In the examined studies, a common case of this is the use of learning styles to help learners receive instructional materials that are most appropriate for them first [21,23,27,28,31,32,34,37,38,42]. Such learning styles can also be modified if the learner's performance falls short of expectations [39]. Felder Silverman's Learning Style (FSLS) [48] model, which categorises learners as active or reflective, global or sequential, visual or verbal, and sensing or intuitive, is a frequent learning style investigated in these works. The FSLS is normally established by a questionnaire that learners must complete prior to beginning the learning process. There are generally arguments for and against using learning styles to increase learning performance in the literature [49,50]. The need for learning styles can be summarised as follows [50]: a. learners will have a preference for their 'style' of learning; b. learners vary in their capacity to study specific information; and c. improved academic outcomes will arise from the 'matching' of curriculum design to learning style of a learner, as characterised by one of the previously stated parameters. The most common criticism against learning styles is that there is little empirical data to support its assertions.

Allowing each learner to progress through the course at their own pace (such as understanding or level of knowledge) is another aspect where performance may be improved. This would entail pedagogical agents organising tutoring activities and personalised learning trajectories based on the grades of past learning sessions of the learners [23].

In terms of better performance, comprehension, memory, and recall, pedagogical agents can give clues and resources related to topics that have been identified to require more emphasis on the learner's part. Such tutoring activities save learners from having to go through the entire learning material, especially if it is huge, and instead focus on the most important parts. González-Castro et al. [41], for example, as part of their design,

include a pedagogical agent that proposes video fragments relevant to ideas for questions that a student does not correctly answer in a MOOC.

### 5.3.2. Task Completion

This expected outcome is particular to a pedagogical agent's function in assisting a student to complete an academic activity within a certain period. This category is intended to distinguish simple duties such as assisting users in getting access into the system or delivering system information.

In our research, an instance emerges where intelligent tutoring tasks such as interactive problem-solving help and interactive solution analysis are performed by agents serving as guides or mentors. Kim [12] states that in such instances, pedagogical agents can mimic the traits of an ideal human mentor, such as professional skill as well as an interpersonal and caring mannerisms. When learners are experiencing difficulty or are unable to solve an issue, pedagogical agents aid them in completing tasks by providing tips or feedback systems [23,24,34,43]. The frequency of such scaffolding, however, is expected to change depending on the learner's competence [7,43], such as level of knowledge [25].

Another case is where the pedagogical agent collaborates by functioning as a virtual student. In this case, the agent operates as a virtual peer and learns with the learner, acting as a coping model for difficult tasks [12]. Throughout the activity, the pedagogical agent engages the learner via voice or text communication, offering important information to help the learner finish the task [25,29,30,36,41].

### 5.3.3. Enhanced Engagement, Motivation, Responsibility

The level of interpersonal connection between peers, but especially between learners and teachers who are to function as guides, mentors, or experts, is one of the fundamental contrasts between e-learning and a classroom-based setting. The situation is more prevalent in an e-learning climate, such as a MOOC [26,41], where there may be thousands of learners and not enough human instructors to go around. As a result, researchers frequently argue that a primary role of incorporating educational agents is to promote motivation, engagement, and responsibility, as the social presence of an agent may be expected to stimulate learners' interest and attention [9,11–14]. Veletsianos and Russell [9] refer to this crucial role as the pedagogical agent's 'persona effect', since the agent's appearance may serve as a social model for the learner, and communication and relationships can be expanded and widened between the learner and the assigned agent, whether as a peer or mentor.

In the evaluated papers, involvement is shown through interactive dialogue in tasks [25,26,29,30,36]. Such discussions are intended to involve the student in the process of working through exercises or addressing challenges. Another engaging approach, according to Bendou et al. [26], is to provide flexibility to the course schedule by allowing extra time to finish assignments. In the model for deaf learners, Hammami et al. [33]'s engaging technique is to modify the learning environment by recognising learning barriers and suggesting educational activities and learning objectives that take the learner's limitation into consideration.

Motivation can be expressed in the form of direct words of encouragement or motivation to the learners, such as the tutor agent in Palomino et al. [22]'s design giving celebratory or encouraging messages based on their performance. Another motivational method is to encourage teamwork with the goal of completing certain tasks [26].

There were few instances in the analysed studies when educational agents instilled responsibility in the students. However, in Duffy and Azevedo [25]'s approach, *Pam*, an agent, aids students in setting proper goals. These objectives are developed from a predetermined set of alternatives. Agents encouraging learners to initiate discussions or acting as learners (teachable agents) in a tutor-learner interaction may also demonstrate responsibility in learners [9].

According to studies, agent-learner connections have the ability to alleviate concerns of loneliness and isolation, hence increasing motivation, engagement, and responsibility [9,12–14,40]. However, because they are linked to emotions, it poses ethical and philosophical problems concerning the depth and appropriateness of the emotional connection between pedagogical agents and learners [9]. This is an area that has received little attention in the literature.

## 6. Conclusions and Future Directions

This study gives a comprehensive assessment of the literature on empirical and conceptual research on pedagogical agents in personalised adaptive learning systems. We focused on the roles that software agents play in such systems from two perspectives: adaptive and intelligent roles. Although adaptivity and intelligence have their origins in adaptive hypermedia and intelligent tutoring systems, respectively, current trends indicate that resilient systems should combine both. This study gives insight on how software agent qualities are employed to compensate for the lack of physical connection in an online setting. The findings of this study can be utilised as a basis for further research into the effects of including pedagogical agents in PALS, particularly on predicted outcomes such as improved performance, task completion, increased motivation, and engagement.

**Author Contributions:** Conceptualisation, investigation, methodology, U.C.A., A.M.A.H. and H.K.M.A.-C.; writing—original draft preparation, U.C.A., A.M.A.H. and H.K.M.A.-C.; writing—review and editing, data curation, U.C.A.; APC Funding acquisition, A.M.A.H., H.K.M.A.-C. and C.B.; supervision, C.B. and M.L.M. All authors have read and agreed to the published version of the manuscript.

**Funding:** This research received no external funding. However, the APC for this publication was partially covered by the Faculty of Automatics, Computer Science and Electronics, University of Craiova, Craiova, Romania.

**Institutional Review Board Statement:** Not applicable.

**Informed Consent Statement:** Not applicable.

**Data Availability Statement:** Not applicable.

**Conflicts of Interest:** The authors declare no conflict of interest.

## Abbreviations

The following abbreviations are used in this manuscript:

| | |
|---|---|
| FSLS | Felder Silverman Learning Style |
| IEEE | Institute of Electrical and Electronics Engineers |
| ITS | Intelligent Tutoring System |
| MAS | Multi-Agent System |
| MOOC | Massive Open Online Course |
| PAL | Personalised Adaptive Learning |
| PALS | Personalised Adaptive Learning System |
| VLE | Virtual Learning Environment |

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
