# Peer review of "The Role of Pedagogical Agents in Personalised Adaptive Learning: A Review"

_sustainability, doi:10.3390/su14116442_

Round 1
Reviewer 1 Report
The manuscript is well written, it adds new information to international literature.
Author Response
The article has been updated, most notably in Section 2 to include related reviews and Section 5.3 to provide a more detailed summary of expected outcomes.
Reviewer 2 Report
The manuscript has a compilation of recent works related to adaptive learning. Two research objectives are addressed: namely defining the responsibilities of pedagogical agents in personalized adaptive learning systems, and their impact in e-learning environments.
The methodology is clearly explained, and while I think the topic is of high relevance, I think the manuscript falls short of achieving the abovementioned objectives, and more importantly so the second of these objectives. I would have expected a deeper analysis of the impact in learners for example from an evidence-based perspective. If that expectation was too high then at least I would have expected a more granular characterization of the impact by, e.g., subject area, or level of education.
Author Response
Section 5.3 has been included to analyse expected outcomes (improved performance, task completion, and Enhanced Engagement, Motivation, Responsibility)
Reviewer 3 Report
The study is an interesting one. I have only one request to incorporate a Prisma diagram, and a pie chart/bar diagram depicting different publications explored (like IEEE, Elsevier, Springer, MDPI etc) if possible.
Author Response
Figure 3, which is a pie-chart, has been included to show the distribution of publishers in the included primary studies
Round 2
Reviewer 2 Report
Section 5.3 does alleviate the lack of comments on the impact of pedagogical agents in personalized adaptive learning. Nonetheless, it still misses the opportunity to provide depth. For example, saying that "improved performance was by far the most represented in the research for the expected outcomes by integrating pedagogical agents (80%)" does not tell us much whether these 80% happened recently or during a certain period of time when perhaps technologically advanced tools became available, or if happened predominantly many years ago when the available technology wasn't as sophisticated.
How about student satisfaction? How about meeting learning outcomes? How about improvements on characterizing students' profiles? Would some graphs help the descriptions?
Does the contents of the manuscript really address "The Role of Pedagogical Agents in Personalised Adaptive Learning"? (BTW, the title need a hyphen or a colon before "A Review"). I believe that without a clear picture of the impact on learners and the lack (or laxed set) of metrics to measure such impact, the manuscript has serious gap.
Author Response
In response to your first comment, I neglected to mention that Section 5.3 also includes 5.3.1, 5.3.2, and 5.3.3, which go into detail about the expected outcomes: improved performance, task completion, increased engagement, motivation, and responsibility. We can add graphs to back up some of those points and include others. But I want to make sure you didn't miss those subsections.
To separate the phrases in the title, a colon has been added.
